# Enhancing Semantic Segmentation with Continual Self-Supervised Pre-training

**Brown Ebouky**                                                       *Brown.Ebouky@ibm.com*
*ETH Zurich*
*IBM Research – Zurich*

**Ajad Chhatkuli**
*INSAIT, Sofia University "St. Kliment Ohridski"*

**Cristiano Malossi**
*IBM Research – Zurich*

**Christoph Studer**
*ETH Zurich*

**Roy Assaf**
*Kaiko*

**Andrea Bartezzaghi**
*IBM Research – Zurich*

**Reviewed on OpenReview:** *https://openreview.net/forum?id=Ax9Y4WOg7s*

## Abstract

Self-supervised learning (SSL) has emerged as a central paradigm for training foundation models by leveraging large-scale unlabeled datasets, often producing representations with strong generalization capabilities. These models are typically pre-trained on general-purpose datasets such as ImageNet and subsequently adapted to various downstream tasks through finetuning. While prior work has investigated parameter-efficient adaptation methods like adapters, LoRA, and prompt tuning, primarily targeting downstream finetuning, extending the SSL pre-training itself in a continual manner to new domains under limited data remains largely underexplored, especially for downstream dense prediction tasks like semantic segmentation. In this work, we address the challenge of adapting vision foundation models to low-data target domains through continual self-supervised pre-training, specifically targeting downstream semantic segmentation. We propose GLARE (Global Local and Regional Enforcement), a novel continual self-supervised pre-training task designed to enhance downstream semantic segmentation performance. GLARE introduces patch-level augmentations to encourage local consistency and incorporates a regional consistency constraint that leverages spatial semantics in the data. For efficient continual pre-training, we initialize Vision Transformers (ViTs) with weights from existing SSL models and update only lightweight adapter modules–specifically UniAdapter–while keeping the rest of the backbone frozen. Experiments across multiple semantic segmentation benchmarks on different domains demonstrate that GLARE consistently improves downstream performance with minimal computational and parameter overhead.

# 1  Introduction

Self-supervised learning (SSL) has revolutionized the training of foundation models by enabling the extraction of rich, generalizable features from vast unlabeled datasets (Caron et al., 2021; Chen et al., 2021; Oquab et al., 2024; Assran et al., 2023). This paradigm has proven to be particularly valuable in computer vision, where the abundance of unlabeled images can be leveraged to learn robust visual representations without the need for expensive manual annotations. Recent advances in SSL have introduced sophisticated techniques, spanning from innovative data augmentation strategies (Grill et al., 2020) and masked image modeling (He et al., 2022) to enhanced feature matching through self-attention mechanisms (Su et al., 2023; Su & Ji, 2024) and refined loss functions.

While these developments have led to powerful models pre-trained on extensive generic datasets like ImageNet (Deng et al., 2009), they often fall short when confronted with specialized technical domains. This limitation becomes particularly apparent in real-world scenarios where domain-specific data is scarce, especially labeled data. The challenge is further compounded when considering dense prediction tasks like semantic segmentation, which demand more fine-grained semantic understanding compared to classification tasks. These issues have therefore sparked significant interest in adaptation techniques, particularly finetuning, and continual learning to bridge the gap between generic pre-training and domain-specific applications.

Our research focuses on adapting a generalist SSL pre-trained backbone to a new domain with limited data through continual pre-training within a pure SSL framework. Unlike continual learning, which typically refers to adapting a model trained for a specific downstream task to a new domain or task, our focus is on the pre-training stage: specifically, we start from a model already pre-trained with SSL on some data and we continue the unsupervised SSL pre-training on new, unlabeled domain data. This allows the model to better align its representations with the target domain using the available unlabeled data prior to the downstream adaptation stage, that then makes use of the scarce available labeled data. Previous work has explored aspects of continual pre-training by, for example, training batch normalization layers using conventional SSL approaches for classification tasks (Reed et al., 2022). However, these studies do not fully investigate other key components of the SSL paradigm, such as augmentations and matching, particularly in limited data regimes. Furthermore, while extensive research exists on continual pre-training for image classification (Lin et al., 2022; Cheng et al., 2023; Reed et al., 2022; Tang et al., 2024), the distinct challenges posed by semantic segmentation warrant separate investigation, as features of foundation models often perform differently between classification and segmentation tasks (Su & Ji, 2024; Caron et al., 2021; Oquab et al., 2024).

Therefore, we focus our work on identifying an appropriate data-efficient technique for the continual SSL pre-training of foundation models with limited unlabeled data, tailored for downstream semantic segmentation. Our key contributions include:

- We explore the benefits of SSL-based continual pre-training for semantic segmentation under limited data conditions. To this aim, we employ a trainable adapter (Lu et al., 2023) after the self-attention layers in ViTs, in order to mitigate catastrophic forgetting during continual SSL pre-training.

- We propose a data-efficient augmentation strategy centered on patch-wise strong blurring, instead of the typical patch-wise or block-wise masking approaches. This seemingly simple modification proves particularly effective in facilitating the learning of inter-patch relationships, especially in limited data settings.

- We propose explicit local and regional consistency enforcement to improve learning dense features with limited data. These constraints help to learn more distinct spatial features crucial for segmentation tasks. Specifically, we propose a regional consistency mechanism by penalizing feature inconsistency between the student (base encoder) and teacher (momentum encoder) within spatial patch groups, processed through a cross-attention layer. In this context, we sample the region features based on attention maps to leverage the semantic content already learned by the original model. Alongside the local patch-wise consistency, the regional consistency helps in learning better spatially distinct features suitable for semantic segmentation.

We provide model-specific experiments with existing state-of-the-art foundation models, while comparing our proposed SSL with standard SSL approaches. We validate our approach through experiments across four datasets: ADE20k (Zhou et al., 2017), Pascal Context (Mottaghi et al., 2014), Cityscapes (Cordts et al., 2016) and LoveDA (Wang et al., 2021a) (satellite images), showing consistent performance improvements on downstream segmentation task after applying the proposed SSL approach in continual pre-training.

## 2 Related Works

### 2.1 SSL for global image understanding

SSL methods for vision aim to learn rich visual representations from large-scale unlabeled data by enforcing various learning objectives and paradigms. One prominent paradigm is joint embedding networks, exemplified by methods like MoCo (He et al., 2020) and DINO (Caron et al., 2021). MoCo employs contrastive learning with a memory bank for gathering negative examples (He et al., 2020; Wu et al., 2018), while later approaches, such as SimCLR (Chen et al., 2020b) and MoCo-V3 (Chen et al., 2021), eliminate the memory bank and use training batch samples as negatives. In contrast, methods like DINO (Caron et al., 2021), BYOL (Grill et al., 2020), and SimSiam (Chen & He, 2021) forego negative pairs altogether, focusing instead on positive pair similarity. Techniques for achieving this include contrastive learning (e.g., ZeroCL (Zhang et al., 2022)), clustering (e.g., SwAV (Caron et al., 2020), SeLa (Asano et al., 2020), MSN (Assran et al., 2022)), and alternative objectives such as redundancy reduction in Barlow Twins (Zbontar et al., 2021).

Another paradigm, inspired by Natural Language Processing (NLP) (Radford et al., 2018; Devlin et al., 2019), is generative SSL. Examples include iGPT (Chen et al., 2020a), which reconstructs masked pixels, and vision-specific approaches like BEiT (Bao et al., 2022) and MAE (He et al., 2022), which reconstruct masked patches. More recently, I-JePa (Assran et al., 2023) advanced this concept by predicting embeddings of masked patches using contextual and target encoders. Hybrid approaches, such as iBoT (Zhou et al., 2022), DINOv2 (Oquab et al., 2024), combine joint embedding learning with masked image modeling techniques like those in MAE (He et al., 2022).

### 2.2 SSL for dense prediction

SSL methods tailored to dense prediction tasks, including semantic segmentation, have gained special attention in recent years. These methods can be grouped into three main categories:

**Learning local features** This group emphasizes learning representations at the pixel or patch level. Methods like PixPro (Xie et al., 2021c) and DenseCL (Wang et al., 2021b) employ contrastive learning to enforce consistency between pixel representations across different views. DetCo (Xie et al., 2021a) integrates instance-patch and patch-level contrastive losses, achieving strong object detection performance without compromising image classification results. LOCA (Caron et al., 2024) clusters matching patch features between query and reference views, encouraging consistent distributions of patch-level clusters. By back-tracking random augmentations (Pinheiro et al., 2020), LOCA establishes patch correspondences. In our approach, we adopt a similar strategy for local consistency, but avoid LOCA's constraints of smaller query images relative to the reference, allowing for richer information. Furthermore, instead of LOCA's clustering approach, we enforce an inter-view local consistency using a DINO-like objective.

**Enhancing spatial awareness** The second category focuses on providing location awareness during pre-training. LOCA (Caron et al., 2024) predicts the positions of query patches within a reference image. Similarly, UP-DETR (Dai et al., 2022) extends the DETR (Carion et al., 2020) framework to localize random patches. Other approaches solve spatial reasoning tasks, such as rearranging jigsaw puzzles (Zhai et al., 2022) or identifying incorrect patch positions (Sameni et al., 2023). More recently, ADCLR (Zhang et al., 2023b) introduces query patch tokens, treating cropped image regions as additional class tokens, enhancing spatial awareness during pre-training.

**Maximizing regional or object-level similarity** This group of methods focuses on similarity within regions or at the object level. ReSim (Xiao et al., 2021) aligns representations of overlapping sliding window

regions across views. At the object level, methods like SoCo (Wei et al., 2021), ORL (Xie et al., 2021b), and SCRL (Roh et al., 2021) employ techniques such as selective search which are computationally expensive. SelfPatch (Yun et al., 2022) defines regions as the set of patches in the direct neighborhood of a central patch and enforces similarity between these neighbors. FLSL (Su et al., 2023) on the other hand, introduces intra- and inter-view clustering, attracting representations of the same concept while repelling clusters of different concepts across augmentations. Finally, the most recent work UDI (Su & Ji, 2024) encourages multimodal local predictions by adding an additional class token and solve the semantic misalignment problem.

In this work, we use UDI pre-trained model to initialize our backbones for continual pre-training as UDI (Su & Ji, 2024) showcases strong results in SSL for dense prediction tasks. We then leverage the best performing model and show consistent improvements using our pre-training strategy in continual pre-training for semantic segmentation.

### 2.3 Continual pre-training

Recent works have explored unsupervised continual pre-training as a means to adapt large-scale pre-trained models to new domains. These approaches aim to bridge the gap between general-purpose pre-training and downstream domain-specific tasks by refining representations learned on source domains. Hierarchical Pre-Training (HPT) (Reed et al., 2022) introduces a framework for self-supervised continual pre-training, where a model trained on source domain data is further pre-trained on target domain data by finetuning all model weights in a sequential framework. Other approaches adapts ViTs using masked image modeling on target domain (Mendieta et al., 2023). More recent efforts have explored parameter-efficient adaptation strategies, such as incorporating lightweight modules like adapters (e.g., LoRA) to reduce compute cost (Scheibenreif et al., 2024; Khanna et al., 2024). While these methods demonstrate the promise of continual pre-training, they are primarily developed for downstream classification tasks and rely on large-scale datasets (typically exceeding 100k images). In contrast, our work focuses on designing a continual self-supervised pre-training pipeline tailored to dense prediction tasks, specifically semantic segmentation, in data-scarce target domains. Rather than proposing a new parameter-efficient tuning method, we repurpose an adapter-based approach and apply it during the continual self-supervised pre-training stage, enabling domain-aligned representation learning prior to downstream adaptation.

## 3 Preliminaries and Problem Formulation

In this section, we describe the learning premise with detailed description of the tools and the problem formulation.

### 3.1 Vision Transformers

Let $X \in \mathbb{R}^{C \times H \times W}$ be an image, where $H$ and $W$ represent the height and width of the image, respectively, and $C$ the number of channels. A Vision Transformer model (ViT) (Dosovitskiy et al., 2021) considers the image $x$ as a set of $N$ non overlapping patches $x_i \in \mathbb{R}^{CP^2}$, of resolution $P \times P$ and with $C$ channels. These patches are then projected through a linear layer to a space of dimension $D$, such that $z_i = W x^{(i)} + E_{pos}^i$, where $W \in \mathbb{R}^{D \times CP^2}$ is a linear projection and $E_{pos}^i \in \mathbb{R}^D$ is the positional embedding for the patch at index $i$. A learnable token $z_{[CLS]} \in \mathbb{R}^D$, referenced as [CLS], is prepended to the sequence of patches to extract global information from the image. The resulting input sequence is thus defined as $z = [z_{[CLS]}, z_1, z_2, \ldots, z_N]$. Then, ViTs take the input to produce global level ($e_{[CLS]}$) and patch level ($e_i$) representations by using its encoder. In the same way as in (Yun et al., 2022; Zhang et al., 2023b), we refer to the encoder as $f_\theta$ with parameters $\theta$ and we use Equation (2) to represent the whole process of a ViT:

$$f_\theta(x) = f_\theta([z_{[CLS]}, z_1, z_2, \ldots, z_N]) \tag{1}$$

$$= [f_\theta^{[CLS]}(x), f_\theta^{(1)}(x), f_\theta^{(2)}(x), \ldots, f_\theta^{(N)}(x)], \tag{2}$$

with $f_\theta^{[CLS]}(x)$ and $f_\theta^{(i)}(x)$ being the final representations of the global image token [CLS] and the $i$-th patch, respectively.

### 3.2  Continual pre-training with adapters

Our interest is to improve the SSL ViT features via a task-agnostic SSL framework in the continual learning setup. Therefore, we limit our work's investigation to finding the right SSL framework for continual learning rather than exploring different continual learning paradigms. Among the various continual learning approaches, adapter-based (Ding et al., 2022; Lu et al., 2023; Hu et al., 2022), and memory or replay based (Winter et al., 2024; Reed et al., 2022; Wang et al., 2024) strategies are considered to be state-of-the-art. However, due to their simplicity and compatibility with a wide range of architectures, adapter-based methods are widely used in continual learning (Scheibenreif et al., 2024; Khanna et al., 2024) and finetuning. We therefore consider a simple adapter named UniAdapter (Lu et al., 2023) composed of two linear layers with activation for adapting features after every self-attention layer in ViT. Therefore, given a pre-trained SSL model, our investigation involves training only the adapter layers while freezing the model weights, following the standard practice. Note that, by using the parameter-efficient adapter used typically for finetuning models, our problem formulation departs from previously used SSL continual learning, for which all network layers are trained (Reed et al., 2022). Our choice is mainly motivated by the need to isolate natural catastrophic forgetting from the suitability of the SSL framework (see  Table 2).

Specifically, in this work we adopt an adapter with a down-projection layer $W_{down} \in \mathbb{R}^{D \times r}$, a nonlinear activation function $\sigma$ (notably ReLu (Agarap, 2018)), and an up-projection layer $W_{up} \in \mathbb{R}^{r \times D}$, where $D$ and $r$ are the embedding and bottleneck dimensions, respectively. The adapter blocks are employed after each attention layer. More specifically, with $x$ being the output of a ViT-block, we have the corresponding output of the adapter defined as:
$$x' = \text{Adapter}(x) = x + s \ \sigma(x W_{down}) W_{up}, \tag{3}$$
where $s > 0$ is a scaling factor.

### 3.3  Problem formulation

Given an encoder trained via SSL (Su & Ji, 2024; Caron et al., 2021), we are interested in improving the output feature embedding by training only the adapter parameters $\theta_A$ via SSL. Given an input image, we denote with $e'$ the output feature embedding of the new ViT with the Adapter. Here $\theta_A$ denotes the trainable parameters of the Adapter layers. Given a new target dataset $\mathcal{T}$, our problem is to study and develop the most suitable SSL framework that maximizes the performance of $e'$. We measure the performance of the feature embedding $e'$ by finetuning our model for semantic segmentation together with an additional decoder. In other words, we use semantic segmentation finetuning as the representative metric for gauging the feature embeddings $e'$. We highlight that this setting differs from traditional supervised continual learning. Here, the *continual* aspect refers to the sequential refinement of the pre-trained weights $\theta_A$ of the ViT backbone on a new unlabeled distribution $\mathcal{T}$.

## 4  Method: GLARE Continual Pre-Training

In this section, we describe our method for pre-training the model parameters $\theta_A$ on the target dataset $\mathcal{T}$ using our proposed SSL strategy.

GLARE (Global Local and Regional Enforcement) is an SSL framework focused on learning representations at different levels during the pre-training process. When we as humans look at a picture, in practice we focus our attention to varying levels of detail (Navon, 1977; Shi et al., 2014). We can summarize these levels as 1) the image as a *whole*, 2) as a set of *regions/objects*, and 3) as a collection of specific details on these regions, which can be encoded into *patches* or *pixels*. GLARE, inspired by this concept, is designed as a three-level pre-training strategy, combining the enforcement of **global**, **regional**, and **local** consistency for learning coarse to fine-grained representations, which are crucial to develop a more in-depth understanding of image data. We adopt the usual student-teacher framework for self-supervised learning described in Caron et al. (2021); He et al. (2020); Chen et al. (2021); Yun et al. (2022); Zhang et al. (2023b). Following the same naming convention as in Chen et al. (2021), we consider two ViT encoders: a *base encoder* $f_{\theta,\theta_A}$ and a *momentum encoder* $f_{\theta',\theta'_A}$, parameterized by $(\theta, \theta_A)$ and $(\theta', \theta'_A)$, respectively. The parameters related to the momentum encoder $(\theta', \theta'_A)$ are updated through an exponential moving average (EMA) of those of the base

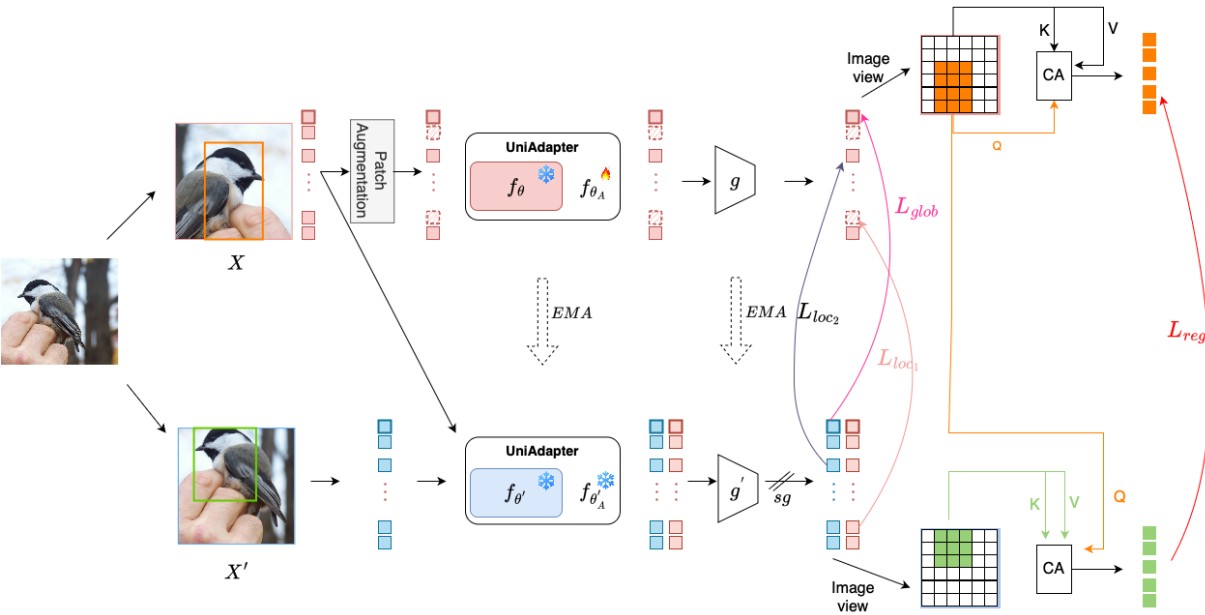

Figure 1: Overview of GLARE continual pre-training framework. Given an image, two views $X$ and $X'$ are generated with image-level augmentation. Each view goes through the base and momentum encoders $f_\theta, \theta_A$ and $f_{\theta', \theta'_A}$. All the parameters are frozen except for those of the adapter on the base encoder. GLARE applies three levels of feature consistency during the pre-training. Firstly, global consistency is considered on [CLS] tokens ($L_{glob}$) of the two views Section 4.1. Secondly, regional consistency is applied on sampled regions, with their representations obtained using a cross-attention module to calculate $L_{reg}$ Section 4.2. Finally, we enforce local consistency focusing on patch-augmentation consistency with distorted vs. not distorted patches of the same view ($L_{loc_1}$) and inter-view local consistency on matching patches from the two views ($L_{loc_2}$). Section 4.3

encoder $(\theta, \theta_A)$, as in He et al. (2020). Figure 1 describes GLARE pre-training strategy. Specifically, in our continual pre-training setup, we train only the adapter parameters $\theta_A$ of the base encoder and update those of the momentum encoder $\theta'_A$ through EMA as stated in Section 3.3. In the following sections, we describe more in detail the three levels of consistency enforced.

## 4.1 Global feature consistency

This pre-training objective focuses on understanding the *overall picture* of an image, also referred to as image-level understanding. Early pre-training methods such as Caron et al. (2021); He et al. (2020); Chen et al. (2021) have tackled this problem using different techniques. The common idea is to have the model learn representations that are invariant to transformations on the image level by maximizing the similarity of representations between augmented views of the same image.

Given an image $I$, a positive pair of views $(X, X')$ is generated by applying random augmentations. In this work, we consider enforcing global consistency by maximizing the similarity of the representations of this positive pair, specifically employing the DINO loss (Caron et al., 2021):

$$L_{glob} = H\left(g_\lambda\left(f_{\theta,\theta_A}^{[CLS]}(x)\right), sg\left(g_{\lambda'}\left(f_{\theta',\theta'_A}^{[CLS]}(x)\right)\right)\right) \tag{4}$$

where $H(a, b) = -a \log b$ is the cross-entropy loss, $sg(\cdot)$ is the stop gradient operation, and $g_\lambda$ is an MLP projection head commonly used in most SSL methods (Caron et al., 2021; Chen et al., 2021; Yun et al., 2022; Zhang et al., 2023b; Su et al., 2023). The parameters $\lambda'$ and $\theta'_A$ are updated with an exponential moving average of $\lambda$ and $\theta_A$.

## 4.2 Regional level consistency

The next pre-training level involves learning region/object representations: we want the model to extract semantic information from regions, whether them being some specific objects (animal, person, etc.) or the background. We aim to enforce the consistency of the representations between two correspondent regions of two views that contain the same semantics. In the context of self-supervised learning, we do not have access to any explicit annotation such as bounding boxes or segmentation masks. We therefore approximate candidate regions through *sampling operations*.

**Region Sampling** This method refers to providing region proposals, similarly to what was done in early object detection models like R-CNN (Girshick et al., 2014), Fast-RCNN (Girshick, 2015), which use selective search to find candidate regions. This process is quite expensive, especially in a self-supervised learning scenario. For that reason, we consider two strategies:

- **Random sampling**: for each region, we randomly sample a starting patch and a number of rows and columns of patches corresponding to the size of our candidate region within an interval $[min_p, max_p]$, where $min_p$, $max_p$ define the minimum and maximum of patches to consider.

- **Attention–aware region sampling**: the attention map of a block of a self-supervised ViT encoder often contain several insights on an image. In fact, over the different heads of the last block, the attention is directed towards different regions, as presented in Figure 2. Consequently, to enforce more semantically-rich regions, we use the attention from the different heads of the encoder to generate the starting patch for candidate regions. In this case, a starting patch is the one getting the most attention on a specific head. From the starting patch, we define the region using a similar process as in *random sampling*. This is feasible primarily in the context of continual pre-training, as it leverages publicly available pre-trained models that already exhibit a useful signal for attention-aware region sampling—something not achievable when training models from scratch.

All results presented in this work employ attention-aware region sampling, as we determined it to be more effective. For more details, see the supplementary material.

**Region correspondence** Let $R$ be a candidate sample region from the view $X$ of the student network, with $z_r$ being the representation of a patch in $R$. By back-tracking the cropping augmentations, we can find the correspondent region $R'$ on the view $X'$ of the teacher network, with patch representations defined as $z_r'$. To encourage the model to learn region-semantic information which aligns with the context, we first extract the semantic context of the query region $R$ with respect to the view $X$, through a cross-attention module:

$$\tilde{z}_r = \mathrm{CA}(z_r, Z, Z) = (W_v Z)\,\mathrm{softmax}(\tau(W_k Z)^T (W_q z_r)), \tag{5}$$

where $W_q$, $W_k$, and $W_v$ are learnable matrices of the cross-attention module, $Z$ the representation of the view $X$ and $\tau$ a scaling factor. We then proceed by enabling semantics sharing which extracts the semantics from the correspondent region $R'$ through the query region $R$, by using the same cross-attention module. This helps the model to extract only relevant information from $R'$, since we know from construction that there are inherent differences between $R$ and $R'$. We obtain the new representation defined as:

$$\tilde{z}_r' = \mathrm{CA}(z_r', Z_R, R) = (W_v Z_R)\,\mathrm{softmax}(\tau(W_k Z_R)^T (W_q z_r')), \tag{6}$$

where $Z_R$ is the representation of the region $R$. Hence, region consistency is enforced with the following loss function applied on the obtained representations $\tilde{z}_r$ and $\tilde{z}_r'$:

$$L_{\mathrm{reg}} = \sum_{\tilde{z}_r \in Z_R, \tilde{z}_r' \in Z_{R'}} H\left(g_\lambda\left(\tilde{z}_r\right), sg\left(g_{\lambda'}\left(\tilde{z}_r'\right)\right)\right). \tag{7}$$

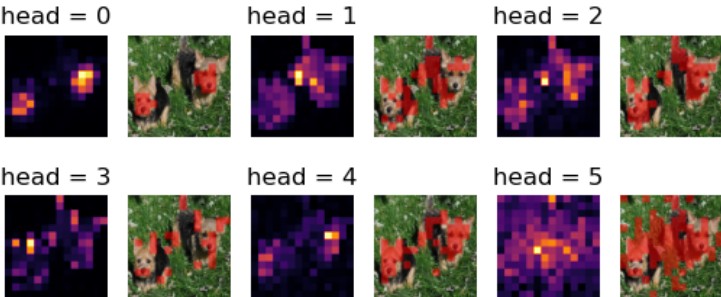

Figure 2: Attention map of the last block of a DINO (Caron et al., 2021) pre-trained model over different heads on an image. The different heads have their attention directed toward specific regions in the image. Some heads focus more on the dog on the left, others on the dog on the right and also on the background.

### 4.3 Local consistency

The final level of pre-training that we consider is the *local consistency*. Under limited data, there are not enough examples to guide SSL to preserve consistency of features between corresponding patches in the base and momentum encoder. We propose and combine two approaches to encourage local consistency. First, we apply random patch blurring, to encourage local patch features to be consistent to each other in the single base encoder view. We call this *patch-augmentation consistency*. Additionally, for all patch features, we also enforce local consistency between the base and momentum encoder. We call this *inter-view consistency*. Note that patch features are the smallest spatial distinction of features in a ViT model. The goal is to extract local semantics across views and between patches within a view to enhance the ability of the model to capture smaller details.

**Patch-augmentation consistency**    Firstly, we aim to alleviate the problem of limited data via separate patch-level augmentations for a fraction of the total image patches. Specifically, we consider strong blurring on 30% of random patches on the base encoder views. This process creates incomplete information on the augmented patches. However, unlike the standard practice in Masked Image Modeling (MIM) (Zhou et al., 2022; Zhang et al., 2023a; Oquab et al., 2024), we do not feed empty tokens for the blurred patches. Thus, we expect the following: *i)* the attention between the local patches in the ViT model are used to "complete" the local patch feature and *ii)* additional scale robustness is imposed on the image patch feature via inter-patch consistency. The main reason for not using MIM in this case, especially for out-of-domain datasets, is that it is known to be data-intensive, requiring a large number of samples and training time to learn meaningful representations (He et al., 2022). This challenge becomes even more pronounced with new domains that exhibit high variability, unlike standard datasets like ImageNet (Deng et al., 2009), and with few samples as typically encountered in continual pre-training. We also investigate other patch-level augmentations such as *rotation* and *noising*, and blurring turns out to be the best-performing approach.

Once, a patch-level augmentation has been defined, we randomly apply it to a set of patches of the view $X$ and we distill the knowledge from the non-distorted patches using the same objective $L_{loc_1}$ as in Zhou et al. (2022); Oquab et al. (2024). Let $X_m$ be a distorted version of the view $X$:

$$L_{loc_1} = \sum_{mk} H\left(g_\lambda\left(f_{\theta,\theta_A}^{[mk]}(x_m)\right), sg\left(g_{\lambda'}\left(f_{\theta',\theta_A'}^{[mk]}(x)\right)\right)\right), \tag{8}$$

with $f_{\theta,\theta_A}^{[mk]}$ corresponding to the mask patches.

**Inter-view local consistency**    Secondly, we learn local semantic information across different views of an image through correspondence. Similarly, to what is done in Section 4.2, we apply a matching algorithm that back-tracks the augmentation process to find correspondence between a patch from the student view $X$, and the ones in the teacher view $X'$. We then proceed by enforcing the consistency between the correspondent patches in the two views $X$ and $X'$. Let $x^{(s)}$ be a patch of the student view $X$ and $C(x^{(s)}) = \{t \in h(x^{(s)} \mid X')\}$

the set of indices of the correspondent patches of $x^{(s)}$ in the teacher view $X'$, with $h(x^{(s)} \mid X')$ being a function which maps a patch in student view to the ones in the teacher by providing the correspondent indices. We can then write the loss function for a given $x^{(s)}$ as:

$$L_{loc_2}^{(s)} = \sum_{t \in C(x^{(s)})} H\left(g_\lambda\left(f_{\theta,\theta_A}^{(s)}(x)\right), sg\left(g_{\lambda'}\left(f_{\theta',\theta_A'}^{(t)}(x')\right)\right)\right), \tag{9}$$

and we then have $L_{loc_2} = \sum_{x^{(s)}} L_{loc_2}^{(s)}$.

Therefore, the local level consistency loss is then defined by $L_{loc} = L_{loc_1} + L_{loc_2}$. Finally, the overall GLARE objective including all pre-training levels is then defined as:

$$L = L_{\text{glob}} + L_{\text{reg}} + L_{\text{loc}}. \tag{10}$$

## 5 Experimental Setup

We use ViT-S/16 (Dosovitskiy et al., 2021) for our experiments due to its cost-efficiency balance between training cost and performance, as in Su & Ji (2024). We apply continual pre-training for 100 epochs on the target dataset, after observing lower performance when training longer. We also used a register token in our ViT encoder as in Darcet et al. (2024); Su & Ji (2024). We use a batch size of 512, and a shared projection head across the different pre-training levels as done in Su & Ji (2024); Zhang et al. (2023b); Zhou et al. (2022) with the output dimension of $K = 8192$. The learning rate is linearly increased for the first epoch to its base value calculated using the linear scaling rule in Chen et al. (2020b), which is $lr = 1.5 \cdot 10^{-4}$. After warmup, the learning rate is decreased using a cosine scheduler (Loshchilov & Hutter, 2017). The weight decay is set to 0.1 and we use AdamW optimizer (Loshchilov & Hutter, 2019). We follow the data augmentations of BYOL (Grill et al., 2020) which was also used in Caron et al. (2021) (i.e. color jittering, gaussian blur, solarization) on random resized crops. Specifically, given an input image we generate 2 global views of resolution $224 \times 224$ and 10 local views of resolution $96 \times 96$ as in DINO (Caron et al., 2021).

We compare GLARE with SOTA SSL methods described in Su et al. (2023); Su & Ji (2024) on ViT-S by analyzing the performances of these pre-training methods against GLARE when they are used from scratch or in a continual pre-training setup. In this work, we initialize our model using the pre-trained weights from UDI (Su & Ji, 2024), which currently represents the state-of-the-art in leveraging self-supervised learning for downstream segmentation tasks. We then train only the parameters of the adapter layer as shown in Section 3.2.

All our experiments are performed employing 8 NVIDIA® A100 GPUs, with 80 GB of memory each.

## 6 Main Results

In the following, we use the notation $ssl_A \rightarrow ssl_B$ to present the results of our experiments. This means that starting from the pre-trained model obtained using the pre-training strategy $ssl_A$ on ImageNet-1k (Deng et al., 2009), we continue the pre-training using the strategy $ssl_B$ on a specific target dataset.

### 6.1 Semantic segmentation performance on different benchmarks

In Table 1, we report the performance on semantic segmentation of models obtained by continual pre-training with different methods starting from UDI weights (Su & Ji, 2024). We also include the performance when the model is randomly initialized without any pre-training knowledge. We report the average mIoU scores and standard deviation after finetuning $3\times$ the pre-trained models on semantic segmentation using FPN (Kirillov et al., 2019). We consider 3 classes of datasets, namely *general domain*: ADE20k (Zhou et al., 2017), Pascal Context (Mottaghi et al., 2014), *driving*: Cityscapes (Cordts et al., 2016) and *aerial*: LoveDA (Wang et al., 2021a), which contain respectively $20k$, 4998, 2975 and 2522 images in their training set, making them suitable for continual pre-training in limited data scenarios ($<= 20k$ images). For each experiment, we first initialize the weights of the backbone with those of the starting model (UDI), then we do continual SSL

Table 1: Comparison of SSL pre-trained models and continual pre-trained models starting from UDI (Su & Ji, 2024) on four semantic segmentation benchmarks. We report mIoU on the validation sets. We use the FPN (Kirillov et al., 2019) framework with 20k iterations and 2k for LoveDA. GLARE continual pre-training from UDI consistently shows improvements over the other pre-training strategies.

| Method | Backbone | ADE20k | P. Context | Cityscapes | LoveDA |
|---|---|---|---|---|---|
| random init. | ViT-S/16 | 10.0 (± 0.03) | 19.0 (± 0.00) | 42.4 (± 0.25) | 29.8 (± 0.04) |
| UDI (Su & Ji, 2024) | ViT-S/16 | 41.2 (± 0.11) | 49.1 (± 0.04) | 74.7 (± 0.01) | 50.9 (± 0.02) |
| UDI → UDI | ViT-S/16 | 41.1 (± 0.11) | 49.2 (± 0.04) | 74.9 (± 0.17) | 51.1 (± 0.01) |
| UDI → FLSL | ViT-S/16 | 41.2 (± 0.06) | 48.7 (± 0.04) | 74.2 (± 0.28) | 49.9 (± 0.12) |
| UDI → GLARE | ViT-S/16 | **41.6** (± 0.13) | **49.3** (± 0.01) | **75.3** (± 0.03) | **51.5** (± 0.01) |

Table 2: Comparison of different continual pre-training framework. Experiments performed starting from UDI (Su & Ji, 2024) pre-trained on ImageNet (Deng et al., 2009). We show the segmentation performance of continual pre-trained models on LoveDA with a finetuning for 2k iterations.

| Framework | Pre-training | mIoU | aAcc | mAcc |
|---|---|---|---|---|
| - | UDI (Su & Ji, 2024) | 50.9 | 70.0 | 63.8 |
| HPT | UDI → UDI | 50.4 | 69.5 | 62.1 |
| | UDI → FLSL | 50.9 | 70.1 | 63.1 |
| | UDI → GLARE | 12.7 | 41.7 | 23.0 |
| UniAdapter | UDI → UDI | 51.1 | 70.1 | 63.9 |
| | UDI → FLSL | 49.0 | 68.5 | 60.9 |
| | UDI → GLARE | **51.5** | **70.2** | **64.3** |

pre-training with an adapter (UniAdapter) on the target dataset by training only the adapter parameters. The obtained model is then used for finetuning for segmentation. We observe that GLARE reports consistent improvement for continual pre-training from UDI weights over all the datasets. On ADE20k, we obtain an improvement of +0.4 over UDI and on LoveDA, which is an out-of-domain dataset with respect to the original pre-training dataset of UDI (ImageNet (Deng et al., 2009)), we achieve an improvement of +0.6. This shows that GLARE is able to take advantage of existing encoded features and new data distribution to improve semantic understanding, which transfer in semantic segmentation.

## 6.2 Comparison with other continual pre-training methods

In this section, we compare two different continual pre-training methods: Hierarchical Pre-Training (HPT) (Reed et al., 2022) and our adapter-based strategy, based on UniAdapter (Lu et al., 2023). We continue the pre-training using these strategies, and then we evaluate the quality of the new features by finetuning on semantic segmentation. We consider LoveDA (Wang et al., 2021a) for this experiment as its out-of-domain nature can help better assess the quality of the strategy. Table 2 presents the results on three metrics: (a) mean intersection over union (mIoU) averaged over all semantic categories, (b) all pixel accuracy (aAcc), and (c) mean class accuracy (mAcc). We observe that overall the adapter-based strategy provides a better improvement for continual SSL compared to HPT. In fact, HPT often results in performance degradation in semantic segmentation, in contrast to what is usually observed for classification tasks.

## 6.3 Influence of the SSL backbone

An important question is the role of the SSL backbone used for the continual pre-training of GLARE. Specifically, we want to examine whether the performance gains observed when starting from UDI also hold

Table 3: Segmentation performance (mIoU) on LoveDA using different SSL backbones with and without continual pre-training. We compare original backbones against continual pre-trained ones with (Reed et al., 2022) and UniAdapter (Lu et al., 2023), using the initial pre-training method vs GLARE. We show results after 2k finetuning iterations.

| Init. | Cont. pre-training | Original | Framework | |
|---|---|---|---|---|
| | | | HPT(31M) | UniAdapter(14M) |
| DINO | → DINO | 50.3 | 28.7 | 28.4 |
| | → GLARE | | | **50.6** |
| FLSL | → FLSL | 50.3 | 50.2 | **50.6** |
| | → GLARE | | | 50.5 |
| UDI | → UDI | 50.9 | 50.4 | 51.1 |
| | → GLARE | | | **51.5** |

Table 4: Evaluation of forgetfulness of our continual pre-trained models. We report the mIoU of the finetuned models (using FPN (Kirillov et al., 2019)) on which we initially performed continual pre-training on LoveDA.

| Pre-training | Pre-training Data | ADE20k | P. Context | Cityscapes |
|---|---|---|---|---|
| UDI (Su & Ji, 2024) | ImageNet (Deng et al., 2009) | 41.1 | **49.2** | 74.7 |
| UDI → UDI | LoveDA (Wang et al., 2021a) | 41.1 | 48.8 | **75.2** |
| UDI → GLARE | | **41.3** | 49.0 | 75.0 |

when initializing from other SSL backbones, in particular FLSL (Su et al., 2023) and DINO (Caron et al., 2021), all pre-trained on ImageNet-1k (Deng et al., 2009). To investigate this, Table 3 reports the results of continual pre-training with GLARE starting from these different backbones, using both HPT (Reed et al., 2022) and UniAdapter (Lu et al., 2023) frameworks. We use LoveDA (Wang et al., 2021a) as the target domain and evaluate performance after finetuning for semantic segmentation. We observe that GLARE consistently improves upon the original models. Moreover, starting from a stronger backbone (e.g., UDI) leads to larger gains. We further hypothesize that the original pre-training strategy influences how effectively GLARE can leverage a given backbone.

### 6.4 Do the continual pre-trained models forget?

In this section, we investigate how much our continual adapter-based pre-trained models forget their previously learned knowledge. In particular, we consider GLARE continual pre-trained model as well as UDI continual pre-trained model on LoveDA. We compare their performances against the original performances of the UDI starting model on the datasets ADE20k (Zhou et al., 2017), Pascal Context (Mottaghi et al., 2014), Cityscapes (Cordts et al., 2016), LoveDA (Wang et al., 2021a). Table 4 reports the finetuning performances. We observe that instead of a decrease in performance relative to the original model, we maintain or outperform it. This suggests that continual pre-training using adapters helps the model to get and maintain more semantic insight from one dataset to another without degrading previous performances.

### 6.5 Results in Classification

While our continual pre-training framework is primarily evaluated on semantic segmentation, it is not limited to this task. To demonstrate its broader applicability, we also apply GLARE continual pre-training to classification. Specifically, we experiment with two out-of-domain datasets: Derm7pt (Kawahara et al., 2019), a dermoscopic benchmark for skin lesion analysis containing ∼800 samples, and COVIDx (Wu et al., 2023), a large-scale x-ray benchmark for COVID-19 detection, where we subsample 20% of the data for continual

Table 5: Top-1 classification accuracy on Derm7pt (Kawahara et al., 2019) and COVIDx (Wu et al., 2023) datasets using a ViT-S backbone. We compare models trained from scratch, DINO (Caron et al., 2021) pre-training, and continual pre-training with DINO → GLARE.

| Dataset | Pre-training | acc@1 (%) |
|---------|--------------|-----------|
| Derm7pt | random init. | 39.6 |
| | DINO (Caron et al., 2021) | 48.0 |
| | DINO → GLARE | **49.1** |
| COVIDx | random init. | 62.5 |
| | DINO (Caron et al., 2021) | 60.7 |
| | DINO → GLARE | **69.2** |

Table 6: Impact of the different pre-training levels in GLARE. We report the mIoU of finetuned continual pre-trained models when trained with different levels of GLARE on 20% of ADE20k.

| Global | Regional | Local | mIoU |
|--------|----------|-------|------|
| ✓ | - | - | 40.9 |
| ✓ | - | ✓ | 41.1 |
| ✓ | ✓ | - | 41.5 |
| - | ✓ | ✓ | 41.1 |
| ✓ | ✓ | ✓ | **41.7** |

pre-training and finetuning to maintain a low-data setting. Models are initialized either randomly, from a DINO (Caron et al., 2021) pre-trained backbone, or from GLARE continual pre-training starting from DINO, and finally finetuning of a classification head. We report in Table 5 the top-1 accuracy of models trained from scratch, initialized from DINO, or from DINO → GLARE. We observe that DINO → GLARE consistently outperforms the other initializations, demonstrating the effectiveness of GLARE continual pre-training even for tasks beyond semantic segmentation.

## 7 Ablation Study

In this section, we investigate the effect of the different components of GLARE and their contribution to its performance. We also provide additional ablations in the supplementary material. Unless stated otherwise, we report the finetuning results of UDI → GLARE on LoveDA using FPN (Kirillov et al., 2019).

### 7.1 Different levels of understanding in GLARE

Our work proposes a pre-training strategy operating at different level of details. In Table 6, we show how global, local, and region understanding interact with each other for downstream semantic segmentation. For this experiment, we run the continual pre-training on 20% of ADE20k and report the performance of the finetuned model when considering some or all of the objectives. We observe that combining all levels of details in the pre-training is crucial for the performance of the continual pre-trained model. In fact, GLARE obtains +0.8 compared to only global consistency pre-training.

### 7.2 Influence of patch-level augmentations

A crucial aspect of GLARE is its ability to learn fine-grained details of the image during the pre-training through local consistency enforcement. As explained in Section 4.3, we introduce patch-level augmentation as a mean to increase local semantics during continual pre-training. In this section, we evaluate patch-level *masking* (typically used in iBoT (Zhou et al., 2022), DINOv2 (Oquab et al., 2024)) and *blurring*. Table 7

Table 7: Ablation of patch-level augmentations. We consider continual pre-training with single or combinations of different patch augmentations on LoveDA. We report the mIoU of the finetuned model on LoveDA using FPN (Kirillov et al., 2019)

| Masking | | Blurring | | mIoU |
|---|---|---|---|---|
| random | block | random | block | |
| - | ✓ | - | - | 50.9 |
| ✓ | - | - | - | 50.9 |
| - | - | ✓ | - | **51.5** |
| - | - | - | ✓ | 51.5 |
| - | - | - | - | 51.3 |

shows the results with the different augmentations when finetuned on LoveDA (Wang et al., 2021a). We experiment with block-wise vs random application of masking and blurring during the pre-training. We consider a prediction ratio $r$ set as 0 with a probability of 0.5 and uniformly sampled from range $[0.1, 0.5]$ as in Zhou et al. (2022). We observe that applying *random blurring* provides the best result and we use it for our continual pre-training setup on LoveDA. We hypothesize that in low-data regimes, blurring serves as a more effective augmentation than stronger perturbations such as masking. Unlike aggressive masking, blurring retains essential information, allowing the model to learn from partially distorted patches while preserving semantic context.

## 8 Conclusion

In this work, we explore continual self-supervised learning, specifically for downstream semantic segmentation. While traditional SSL methods are effective for general-purpose pre-training, we find that they struggle to adapt to new domains when used for continual pre-training, particularly on out-of-domain datasets. To address this, we use an adapter for efficient knowledge transfer and propose GLARE, an SSL framework that learns representations at multiple levels: (i) global consistency at the image level, (ii) regional consistency via attention-based candidate regions, and (iii) local consistency through patch-wise augmentation and inter-view patch consistency. This multi-level approach equips the continual pre-trained model with semantically rich representations that improve transferability for segmentation. Experiments on diverse datasets, including both general and out-of-domain (satellite) images, demonstrate GLARE's effectiveness in continual pre-training for semantic segmentation. Our findings advance continual SSL for dense prediction tasks and offer practical insights for adapting foundation models to specialized domains.

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

# A   Details About Datasets

In this paper, we perform continual learning and report results on four semantic segmentation datasets: ADE20k (Zhou et al., 2017), Pascal Context (P. Context) (Mottaghi et al., 2014), Cityscapes (Citys.) (Cordts et al., 2016) and LoveDA (Wang et al., 2021a). We use the training set of these datasets to apply our continual learning setup with different pre-training methods such as UDI (Su & Ji, 2024), FLSL (Su et al., 2023), GLARE. We then finetune the resulting models for segmentation using FPN (Kirillov et al., 2019) and report the performances of the models on the validation sets. In the following we provide more details on these datasets.

**ADE20k (Zhou et al., 2017)**. This dataset contains various scenes which are potentially cluttered with many objects. It includes fine-grained labels with 150 semantic classes. The training set is composed of 20,210 images and the validation set contains 2,000 images.

**Pascal Context (Mottaghi et al., 2014)**. This is a segmentation dataset with denser annotations, which includes background classes like sky and grass in addition to foreground objects. The training set is composed of 4,998 images and the validation set contains 5,105 images. This dataset has 60 semantic classes.

**Cityscapes (Cordts et al., 2016)**. This dataset is designed specifically for urban street scenes, showcasing objects in driving scenarios. It includes 19 semantic categories for segmentation. Its training set is composed of 2,975 images and its validation set is composed of 500 images.

**LoveDA (Wang et al., 2021a)**. This focuses on different geographical environments between urban and rural. It contains high spatial resolution (0.3m) remote sensing images containing objects at different scales, complex backgrounds and inconsistent class distributions. Its training set is composed of 2,522 images, its validation set is composed of 1,669 images and a test set of 1,796 images.

**Derm7pt (Kawahara et al., 2019)**. This is a dermoscopic benchmark for skin lesion analysis containing ∼800 images. The dataset focuses on the classification of pigmented skin lesions into seven diagnostic categories and is commonly used to evaluate models in low-data medical imaging scenarios.

**COVIDx (Wu et al., 2023)**. This is a large-scale chest X-ray benchmark designed for COVID-19 detection. It contains images labeled into three categories: normal, pneumonia, and COVID-19. The dataset aggregates X-ray scans from multiple sources and institutions, making it a diverse benchmark for evaluating models in medical image classification tasks.

Moreover, these datasets differ between each other by their size and their domain. Indeed, by evaluating our continual pre-training on these datasets we showcase the ability of our setup to work in scenarios with limited data and also out-of-domain with respect to the dataset used to pre-train the initial weights (especially in the case of LoveDA). This also stems the contrast between the dataset sizes that we consider in our setup ($<= 20k$) compared to what is used for standard pre-training which are in the order of millions of images.

# B   Other Experimental Details

## B.1   Image-level data augmentation

The augmentation settings in GLARE are based on the augmentation pipeline of BYOL (Grill et al., 2020). In our approach, we begin by sampling two random crops from the input image using a large crop ratio (e.g., $0.25 \sim 1.0$) of size $224 \times 224$. We then proceed by sampling 10 other crops with a smaller crop ratio (e.g., $0.05 \sim 0.25$) of size $96 \times 96$. We use an asymmetric training process where the larger crops, usually referred to as global crops, are passed to the momentum encoder and then all crops (both global and local, which are the smaller ones) are passed to the base encoder. The distortions that we apply are:

- color jittering, with a probability of 0.8, brightness of 0.4, contrast of 0.4, saturation of 0.2 and hue of 0.1;

Table 8: Time requirements of continual pre-training.

| Method | $T_{100}$ |
|---|---|
| UDI $\to$ UDI | 27 min |
| UDI $\to$ FLSL | 18 min |
| UDI $\to$ GLARE | 30 min |

- gray scaling, with a probability of 0.2, gaussian blurring and solarization with probabilities of $(1.0, 0.0)$, $(0.1, 0.2)$ and $(0.5, 0.0)$ for the first, second global crops and the local crops, respectively;

- color normalization, with mean $(0.485, 0.456, 0.406)$ and std. dev. $(0.229, 0.224, 0.225)$.

### B.2 Evaluation details

For evaluation, we perform semantic segmentation on our four segmentation datasets, described in Appendix A. We follow the configurations of the package *mmsegmentation*[1] for finetuning, within the FPN (Kirillov et al., 2019) framework. We consider two configurations for the finetuning: we use 20k iterations schedule for all datasets except for LoveDA for which we use 2k iterations schedule. The resolution of input images during the experiments is $512 \times 512$. Then, as performance metrics we calculate the mean intersection over union (mIoU), the all pixel accuracy (aAcc) and the mean class accuracy (mAcc), for each dataset after finetuning.

### B.3 Computation time

Table 8 reports the continual pre-training time required of three configurations UDI $\to$ UDI, UDI $\to$ FLSL, and UDI $\to$ GLARE on 100 epochs $T_{100}$. The experiments are done on LoveDA (Wang et al., 2021a). We observe that GLARE continual pre-training has higher requirement in terms of computation time. Nevertheless, the continual pre-training is still relatively fast (30 min) since we are only training the adapter parameters for continual pre-training. As for downstream segmentation finetuning, the computation requirement is the same among all the configurations.

## C Additional Ablations

### C.1 Ablation of region sampling

One of the advantages of GLARE continuous pre-training pipeline is its ability to benefit from previously pre-trained models by leveraging learned semantics. This is done for example with region-level understanding which leverages the attention of the pre-trained model to guide region consistency enforcement. In this section, we ablate the use of attention to guide the region sampling compared to random region sampling: *attention-aware region sampling* and *random sampling*. Table 9 presents the results of finetuning the continual pre-trained model on LoveDA on either of these strategies. We experiment with 3 and 6 randomly sampled regions and with 6 regions sampled using attention-awareness. We observe that attention-aware sampling shows an improvement of +0.29% compared to random sampling, which aligns with the hypothesis of having more semantically meaningful regions using the attention map.

### C.2 Effect of blurring strategy

In this section, we study how the blurring is applied during the pre-training process. There are two possible strategies which can be used: *random* and *block-wise* blurring applied on the patches. This is similar to what can be done in masking (Zhou et al., 2022). Table 10 presents the results of models undergone continual pre-training with GLARE using these two different strategies and finetuned on semantic segmentation. We use

---

[1]https://github.com/open-mmlab/mmsegmentation

Table 9: Ablation of the region sampling strategy. Experiments performed with UDI → GLARE with a finetuning of 2k iterations on LoveDA. $M$ represents the number of sampled regions considered.

| Strategy | mIoU | aAcc | mAcc |
|---|---|---|---|
| random ($M = 3$) | 51.3 | 70.0 | 63.8 |
| random ($M = 6$) | 51.4 | 70.2 | 64.0 |
| attention-aware ($M = 6$) | 51.5 | 70.3 | 64.3 |

Table 10: Ablation of the block-wise vs. random blurring. Experiments performed with UDI → GLARE with a finetuning of 2k iterations on LoveDA.

| Method | mIoU | aAcc | mAcc |
|---|---|---|---|
| random initialization | 19.3 | 37.4 | 34.4 |
| block-wise blurring | 51.5 | 70.2 | 64.1 |
| random blurring | 51.5 | 70.2 | 64.3 |

LoveDA as our reference dataset. We observe that applying *random blurring* leads to the best performance. Therefore, we decided to use that strategy for our main experiments in this work, unless stated otherwise.

### C.3 Effect of dataset scale

In this section, we evaluate the performance of our continual pre-training pipeline across different dataset scales. Specifically, we conduct experiments on ADE20K and LoveDA using subsets of 10%, 20%, 50% and 100% of the data. The results are summarized in Figure 3. We observe that even with a small dataset of 2k images (corresponding to 100% of LoveDA and 10% of ADE20K), our continual pre-training approach yields improvements. However, for highly out-of-domain datasets like LoveDA, we hypothesize that performing continual pre-training on a smaller set of unlabeled data can be detrimental. In contrast, ADE20K, which consists of images more closely aligned with ImageNet-1K, does not exhibit the same issue.

## D Visualization of Attention Maps

In this section we visualize some attention maps from the last block of the ViT encoder, using the [CLS] token as the query token. Figure 4 shows the attention from DINO (Caron et al., 2021), the original pre-trained weights from UDI, and a GLARE continual pre-trained model starting from UDI (Su & Ji, 2024) weights, finetuned on ADE20k. We observe similarities in how the attention is distributed across the images, focusing on various details such as foreground objects, object parts, and the background. GLARE continual pre-trained model demonstrates reduced noise relative to UDI, with its attention more precisely directed toward specific objects or regions. When using GLARE for continual pre-training, the model leverages what has been learned before and learns supplementary semantics specific to the dataset.

## E PCA Visualization of Embeddings

In this section, we provide an additional qualitative analysis of the learned representations through PCA visualizations of the embeddings. Specifically, we project the patch representations into three principal components using PCA and visualize them in RGB space. We compare DINO (Caron et al., 2021) pre-trained model against our continual pre-trained variant, DINO → GLARE. As shown in Figure 5, GLARE produces less noisy and more semantically coherent embeddings on both the COVIDx (Wu et al., 2023) and Derm7pt (Kawahara et al., 2019) images.

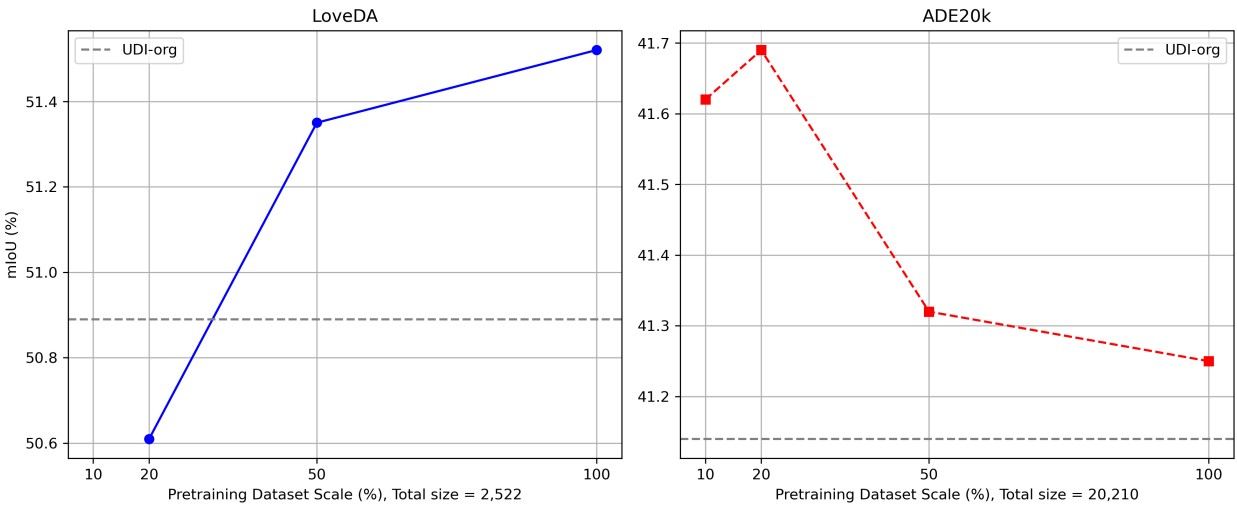

Figure 3: Effect of dataset scale on the performance of GLARE continual pre-training applied on LoveDA and ADE20k. The dashed gray line represent the baseline performance of UDI pre-trained model on the respective dataset.

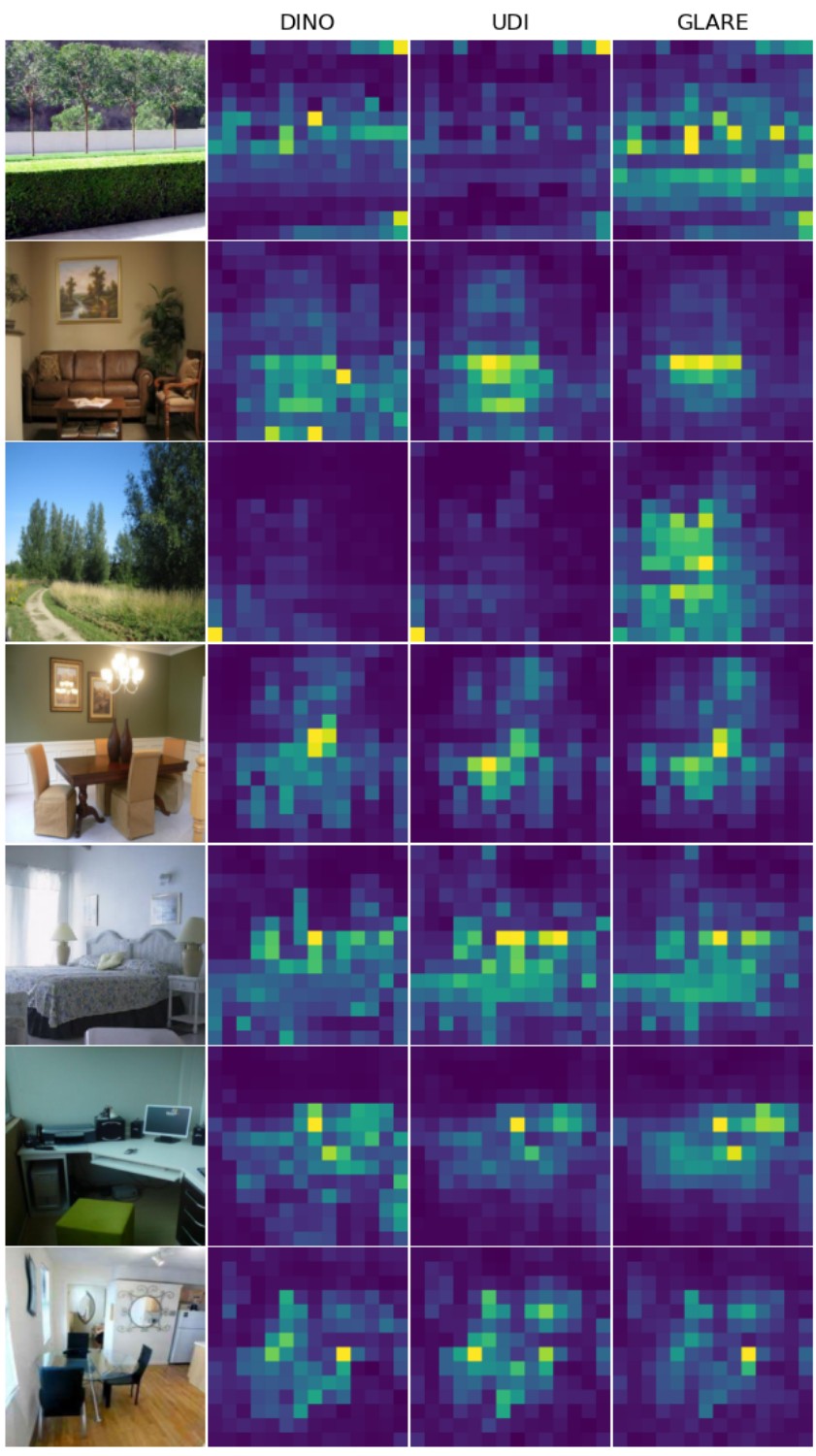

Figure 4: Visualization of self-attention maps obtained from DINO, UDI and GLARE continual pre-trained models from the last block of the ViT encoder starting from UDI.

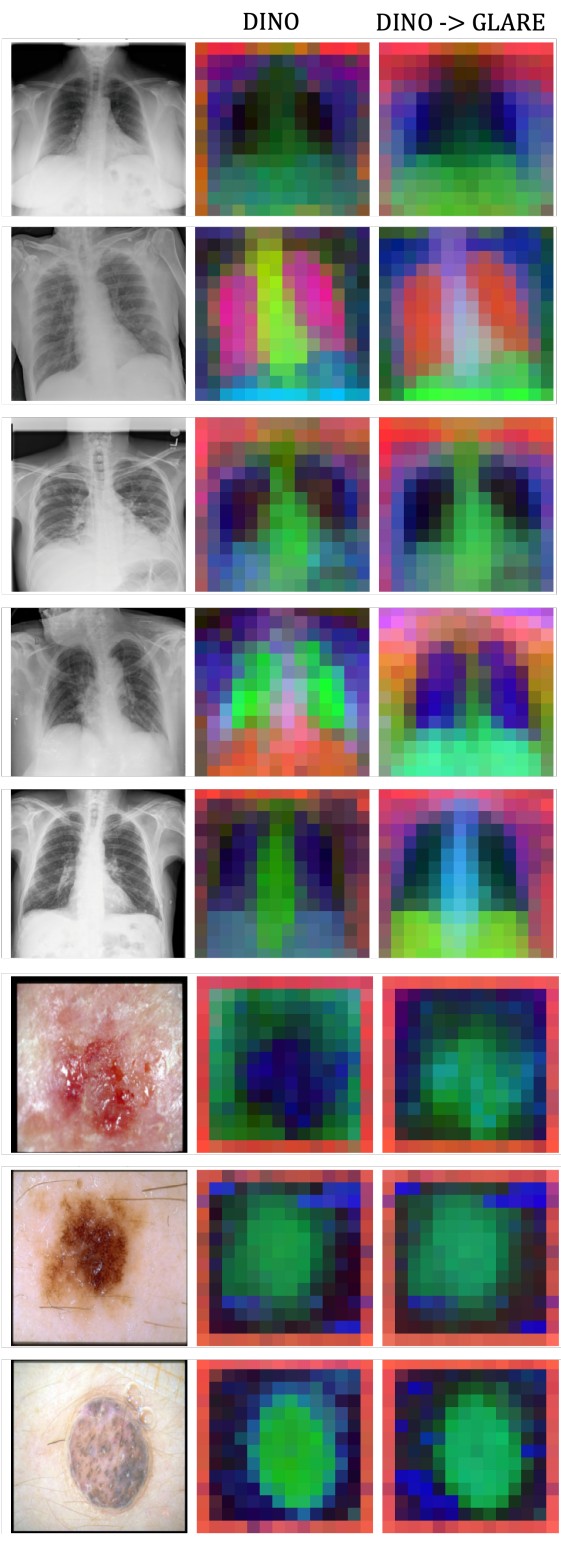

Figure 5: PCA Visualizations of the embeddings on images from COVIDx (Wu et al., 2023) and Derm7pt (Kawahara et al., 2019) datasets. We take the first 3 principal components and show the results for DINO and DINO → GLARE continual pre-trained model.

