# OpenReview forum: "Enhancing Semantic Segmentation with Continual Self-Supervised Pre-training"
_TMLR — Accepted by TMLR_

### Review · Reviewer_Wsh1 · 2025-07-23

**Summary Of Contributions:**

This paper proposes a self-supervised adaptation method for downstream semantic segmentation tasks without fine-tuning the backbone foundation model. The adaptation process involves several modules, including global consistency, region correspondence, and local alignment. The design of these modules appears reasonable, and the introduction of the region correspondence module is particularly novel. Experimental results demonstrate performance improvements across multiple benchmark datasets when these modules are applied.

**Audience:**

Yes

**Audience Explanation:**

The problem this paper aims to address is indeed important, especially as foundation models become increasingly mainstream for downstream applications. Developing an efficient and effective adaptation method for downstream tasks is crucial when leveraging foundation models.

**Claims And Evidence:**

No

**Claims Explanation:**

The paper claims that GLARE consistently improves downstream performance with minimal computational and parameter overhead. However, this claim is difficult to verify, as the experiments do not include comparisons with existing methods in terms of efficiency or resource usage.

**Requested Changes:**

1. The paper is motivated by continual self-supervised pre-training, but the notion of "continual" is not clearly demonstrated. It remains unclear how the proposed method differs from standard (non-continual) pre-training, and how the continual aspect contributes to the final performance.

2. Although the three modules are proposed and combined to improve performance, the gains appear marginal (e.g., +0.6 with region correspondence, +0.2 with local alignment). Given the added complexity of these modules, the trade-off between performance improvement and efficiency is not well justified.

3. The experimental evaluation lacks comparisons with existing methods. The results are only benchmarked against a baseline backbone, which limits the assessment of the method’s relative effectiveness.

---

> ### Author Response · Authors · 2025-09-11
> **Response to Reviewer Wsh1**
>
> We sincerely thank the reviewer for their careful reading of our paper and for the constructive feedback. We have carefully considered the raised concerns and will use them to improve both the clarity of our manuscript and the scope of our experimental evaluation. Below, we address each of the main points raised.
>
> 1. Regarding the notion of **“continual” pre-training**: In GLARE, the continual aspect lies in explicitly reusing knowledge from an already pre-trained model. Specifically, we leverage the semantic information contained in the model’s attention maps to generate candidate regions, on which we enforce region consistency. This mechanism directly reuses previously learned representations and integrates them into the next stage of pre-training, rather than ignoring prior knowledge. In this way, GLARE is different from standard pre-training approaches.
>
>
> 2. While the performance gains of individual modules may appear modest, each module conceptually addresses a complementary aspect of the learned representations — region consistency, local alignment, and global consistency; collectively, they enhance the richness of learned representations which is important for the downstream tasks.
>
> Moreover, we highlight that our pipeline with GLARE is parameter-efficient: as shown in the table below, GLARE combined with UniAdapter requires substantially fewer trainable parameters than HPT, while still improving performance. This demonstrates a favorable trade-off between efficiency and effectiveness.
>
> 3. Finally, the reviewer raised an important point about comparisons with existing methods. We have expanded our evaluation to include continual pre-training of different SSL backbones using different continual pre-training frameworks, namely HPT (Reed et al., 2022) and UniAdapter (Lu et al., 2023). The updated results (Table below) show that GLARE pre-training applied on top of other
> pre-trained models (DINO, FLSL, UDI) achieves improvements over their counterpart, except for FLSL → FLSL.
>
>
> Table. Segmentation performance (mIoU) on LoveDA using different SSL backbones with and without continual pre-training. We compare original backbones against continual pre-trained ones with HPT and UniAdapter, using the initial pre-training method vs GLARE. Results are reported after 2k finetuning iterations.
>
> | Init. | Cont. pre-training | # Train. params. | Original | HPT  | UniAdapter |
> | ----- | ------------------ | ---------------- | -------- | ---- | ---------- |
> | DINO  | → DINO             | 31M              | 50.3     | 28.7 | 28.4       |
> |       | → GLARE            | 14M              |          |      | **50.6**   |
> | FLSL  | → FLSL             | 31M              | 50.3     | 50.2 | **50.6**   |
> |       | → GLARE            | 14M              |          |      | 50.5       |
> | UDI   | → UDI              | 31M              | 50.9     | 50.4 | 51.1       |
> |       | → GLARE            | 14M              |          |      | **51.5**   |

---

### Review · Reviewer_zuJ4 · 2025-07-31

**Summary Of Contributions:**

In this paper, GLARE, a self-supervised method for pre-training, was proposed to learn learning representations at different levels, which is inspired by how human look at a picture and can enhance downstream segmentation performance. The authors specifically target segmentations in data-scarce domains.

Strengths:
1. The idea of learning feature representations at global, regional and local levels seems reasonable.
2. Experiments were conducted on multiple datasets varying in scale and domain.

Weaknesses:
1. The motivation for focusing specifically on segmentation in data-scarce domains is unclear, given that the proposed method appears generally applicable. Would it not also perform well on larger datasets or other tasks such as image classification? Furthermore, could the method---excluding the use of LoRA adapters---not be applied to pretrain from scratch?
2. Is it accurate to characterize ADE20K---a dataset with 20K images---as a small dataset and use it to demonstrate the use of the proposed method in low-data regime? If the authors believe the proposed method is broadly applicable, it would be valuable to evaluate it on medical imaging datasets, which are typically smaller in size and would better support the method’s claimed data efficiency.
3. While well motivated, the performance gain seems very limited.
4. It is mentioned that (1) data augmentation methods like color jittering, gaussian blur,  and solarization are used, and (2) the corresponding region $R^{\prime}$ on the view $X^{\prime}$ is found by back-tracking the augmentations. However, are the data augmentation methods ``backtrackable''?

**Audience:**

Yes

**Audience Explanation:**

The paper addresses vision model pretraining, which is highly relevant to researchers in machine learning, especially those working on transfer learning, domain adaptation, and representation learning.

**Claims And Evidence:**

No

**Claims Explanation:**

1. The authors aim to adapt vision foundation models to new domains in an unsupervised and data-efficient manner. However, they use UDI, a model that is relatively unknown in the field and may not qualify as a widely accepted vision foundation model.
2. The claim that “parameter-efficient strategies for adapting pre-trained models to limited data regimes and for dense prediction tasks remain underexplored” is not well supported. In fact, one of the primary motivations for using pre-trained models is to enhance performance in low-data settings, and substantial research has already been devoted to parameter-efficient adaptation methods---such as adapters and LoRA---across both classification and dense prediction tasks. A clearer justification or positioning relative to existing work would strengthen the paper’s framing.
3. UDI appears to perform well on the target datasets even without continual pretraining, which raises the question of whether the proposed approach provides a meaningful advantage over existing capabilities.
4. The performance improvements reported are limited. Additionally, the authors did not report the segmentation performance on each test dataset when training a segmentation model from scratch---that is, without any pretraining or continual pretraining.

**Requested Changes:**

1. The paper refers to UDI as a vision foundation model, but this model is relatively unknown in the field. Please clarify why UDI qualifies as a vision foundation model and justify its selection over more established alternatives (e.g., DINO, MAE, SAM, or CLIP). If UDI is not widely recognized, framing it as a foundation model may be misleading.

2. The authors state that parameter-efficient adaptation for dense prediction and low-data regimes is underexplored, but this claim is not well substantiated. Please revise this framing to more clearly distinguish your unique contribution from existing literature, particularly in light of substantial prior work on adapters, LoRA, and prompt tuning in similar settings.

3. The method appears generally applicable beyond segmentation in low-data regimes. Please consider including experiments on: (a) different tasks such as classification, and (b) genuinely small datasets (e.g., from medical imaging) to validate the method’s data efficiency.

4. Please include segmentation results from models trained from scratch (without any pretraining or continual pretraining) on each test dataset. This would help quantify the actual gains introduced by your method.

5. The paper mentions backtracking augmentations (e.g., color jittering, Gaussian blur, solarization) to find corresponding regions between views. However, these augmentations are typically non-invertible or non-deterministic. Please clarify: (a) how this backtracking is done, (b) which augmentations are assumed to be reversible, and (c) how approximation errors (if any) are handled in practice.

6. You use ADE20K for demonstrating the use of the method in low-data regime, yet it contains 20K images. Please provide a clearer definition of what constitutes a low-data regime in your context

7. The overall reported performance gains are limited. To better assess the method’s value, please analyze where the method brings the most benefit.

---

> ### Author Response · Authors · 2025-09-11
> **Response to Reviewer zuJ4**
>
> We want to thank the reviewer zuJ4 for taking the time to review our work and for the detailed and constructive feedback. We address here each point that were raised and we will incorporate these additional insights in the final version of the manuscript if given the chance:
>
> 1. **UDI as a vision foundation model**: We agree that UDI is less well known compared to widely adopted models such as DINO, MAE, or CLIP. Our choice of UDI was motivated by its state-of-the-art performance particularly on downstream segmentation among image-only self-supervised learning methods on ImageNet-1k, at the time of our study, as well as its ViT-based architecture, and large-scale training setup. These characteristics align with the definition of a vision foundation model, even if its visibility in the community is more limited.
>
>    We also note that our framework is not tied to UDI specifically: results with DINO and FLSL are included in Table 3, where GLARE continual pre-training, for example with DINO, improves performance on LoveDA. Models such as SAM or CLIP, while strong in their respective domains, are less appropriate baselines here as they are not purely self-supervised image models: SAM is trained for segmentation with dense supervision, while CLIP is trained for image–text alignment. Our focus is on image-only SSL-based vision backbones, which makes UDI and DINO suitable for evaluation.
>
> 2. **Parameter-efficient adaptation for dense prediction and low-data regimes underexplored:** Our claim does not intend to dismiss prior work on adapters, LoRA, or prompt tuning, which have indeed been extensively studied. Rather, our contribution is to highlight that most existing parameter-efficient approaches are applied directly to downstream tasks (e.g., segmentation, classification, detection). In contrast, our work investigates the use of parameter-efficient approaches targeting the continual self-supervised pre-training aspect in new domains, particularly under low-data regimes and for downstream dense prediction tasks. To the best of our knowledge, this combination – efficient unsupervised domain adaptation via continual SSL pre-training, specifically tailored for segmentation – remains underexplored, compared to the well-studied paradigm of directly applying PEFT to downstream supervised fine-tuning.
>
> 3. **Results beyond segmentation & other datasets**: While we agree that the proposed method could, in principle, be applied to tasks such as classification, our work is intentionally focused on downstream semantic segmentation as it represents a challenging prediction task where continual SSL pre-training has been underexplored, particularly under limited data availability. As an additional experiment, we have run continual pre-training on two small datasets targeting **downstream classification**. Specifically, we considered the **Derm7pt** dataset, a dermoscopic benchmark for skin lesion analysis, composed of 800 samples, and **COVIDx** dataset, which is a large-scale x-ray benchmark designed for COVID-19 detection with 3 classes (“normal”, “pneumonia”, “COVID-19”). Regarding the COVIDx dataset, we subsample it by taking 20% of the dataset for continual pre-training and finetuning, equivalent to 4,000 images. We consider a ViT-S backbone and compare models trained from scratch, pre-trained with DINO or continually pre-trained with GLARE, starting from DINO. We freeze the backbone and only finetune a classification head on top (linear probling). We report in the table below the acc@1 metric obtained on the different datasets. Results highlight how continually pre-training with GLARE improves performance in both cases.
>
> | Dataset | Pre-training | Acc\@1 (%) |
> | ------- | ------------ | ---------- |
> | Derm7pt | From scratch | 39.6       |
> |         | DINO         | 48.0       |
> |         | DINO → GLARE | **49.1**   |
> | COVIDx  | From scratch | 62.5       |
> |         | DINO         | 60.7       |
> |         | DINO → GLARE | **69.2**   |

---

> ### Author Response · Authors · 2025-09-11
> **Response Reviewer zuJ4 (end)**
>
> 4. **Results from models trained from scratch:** Following the request, we have added results for models trained from scratch (i.e., without any pre-training or continual pre-training) under the same fine-tuning configurations. These results are now included in Table 1 of the revised manuscript. As expected, models trained from scratch perform substantially worse than those initialized with
> pre-trained weights, thereby highlighting the importance of pre-training. We provide Table 1 as well below:
>
> | Method              | Backbone  | ADE20k (mIoU)     | P.Cont (mIoU)     | Cityscapes (mIoU) | LoveDA (mIoU)   |
> |---------------------|-----------|-------------------|-------------------|-------------------|-----------------|
> | *random init.*      | ViT-S/16  | 10.0 (±0.03)      | 19.0 (±0.00)      | 42.4 (±0.25)      | 29.8 (±0.04)    |
> | UDI                 | ViT-S/16  | 41.2 (±0.11)      | 49.1 (±0.04)      | 74.7 (±0.01)      | 50.9 (±0.02)    |
> | UDI → UDI           | ViT-S/16  | 41.1 (±0.11)      | 49.2 (±0.04)      | 74.9 (±0.17)      | 51.1 (±0.01)    |
> | UDI → FLSL          | ViT-S/16  | 41.2 (±0.06)      | 48.7 (±0.04)      | 74.2 (±0.28)      | 49.9 (±0.12)    |
> | **UDI → GLARE**     | ViT-S/16  | **41.6 (±0.13)**  | **49.3 (±0.01)**  | **75.3 (±0.03)**  | **51.5 (±0.01)**|
>
>
> 5. (a) In our method, the backtracking refers specifically to the cropping augmentation. During training, we record the crop parameters applied to each image. Using these parameters, we map a region from a cropped view back to the original image coordinates, and then locate the corresponding region in another crop of the same image. (b) We do not attempt to invert color-based augmentations such as color jittering, Gaussian blur, or solarization. These transformations are treated as appearance distortions, and the model is explicitly encouraged to be invariant to them. (c) Since our correspondences are established solely from recorded crop parameters, the only source of approximation stems from discretization effects when mapping pixel coordinates across crops. These errors are minor and have no practical impact on the training.
>
>
> 6. **Low-data regime**: In our work, the term low-data regime is used specifically in the context of self-supervised learning. Standard SSL approaches are typically trained on datasets containing millions to hundreds of millions of images (e.g., ImageNet-1k, ImageNet-21k, LAION). By contrast, our continual self-supervised pre-training is performed on datasets of ≤ 20k images, such as ADE20K. While 20k images may not appear small in absolute terms, it is orders of magnitude smaller than the scales usually considered sufficient for effective SSL, and therefore constitutes a low-data setting in this context.
>
>
> 7. **Overall reported performance gains:** While the absolute gains may appear modest, it is important to emphasize that semantic segmentation in low-data and domain-shift settings is a highly challenging problem, where even small improvements are valuable. Our contribution is not in modifying the segmentation pipeline itself, but in strengthening the feature representations through efficient continual self-supervised pre-training. The benefit of our method lies in producing features that transfer more effectively to new domains, leading to consistent improvements across benchmarks.

---

### Review · Reviewer_KXij · 2025-08-28

**Summary Of Contributions:**

## Paper Summary:
This work proposes GLARE (Global Local and Regional Enforcement), a continual self-supervised learning method that enhances the representations of pre-trained foundation models for semantic segmentation. The approach integrates UniAdapters into a DINO ViT backbone and applies a three-level consistency objective covering global, regional, and local representations. The framework is evaluated by fine-tuning on ADE20k, Pascal Context, Cityscapes, and LoveDA, yielding improvements compared to the baseline approach.


## Strengths:

### Relevant research problem:
The paper tackles the improvement of dense feature representations in self-supervised pre-trained models, which is an important and relevant research direction.

**Additional Comments:**

To be fully transparent, I already reviewed this paper for a recent major computer vision conference. The authors received three constructive and detailed reviews. Comparing my previous review with the current submission, it seems that only a few minor writing issues were addressed, while the paper was largely copied and pasted into the TMLR template without addressing any major concerns raised by the reviewers. In practice, three reviewers invested significant time providing helpful feedback, which at this stage appears to have been largely ignored. Unless at least some of my concerns are addressed in a revision, I refuse to recommend this work for acceptance.

**Audience:**

No

**Audience Explanation:**

### Vague motivation and task definition:

The motivation for self-supervised fine-tuning of pre-trained self-supervised representations on a target domain remains unclear. Moreover, the goals of continual learning and pre-training contradict each other, which is not addressed in the paper. Additionally, the continuous aspect of the method remains unclear. While improving the dense feature representation of self-supervised features is a relevant research direction, the goal should be to obtain general representations that generalize across different domains.


### Room for improvement in writing and formatting:

The paper would benefit from a clearer structure. The paper formatting, particularly for figures and tables, suffers from copy-pasting from a two-column template into the single-column template without much adjustment. Tables and figures are not top-aligned, and the references are inconsistent: some include URLs while others do not, and conference venues are formatted inconsistently even across the same venue. Additionally, the method figure (fig. 1) is confusing and hard to follow.

**Claims And Evidence:**

No

**Claims Explanation:**

### Unconvincing quantitative results and limited evaluation scope:

The proposed method does not significantly improve feature representation and is only compared to a single baseline using a single architecture. Compared to the baseline, the method leads to an improvement of about 0.5% mIoU after fine-tuning the enhanced features and a task head for semantic segmentation. The evaluation focuses on supervised probing for semantic segmentation and the ViT-S/16 architecture. Qualitative evaluation is also missing (e.g., PCA visualization of the features). None of the provided results would justify the additional training overhead compared to the initial self-supervised pre-training. More importantly, this work builds up on self-supervised DINO features of a ViT-S/16, arguably a quite weak model for dense prediction tasks like semantic segmentation, as the feature dimension is small, the spatial resolution is low, and the representations are weak compared to more recent approaches such as DINOv2 [A]. Experiments with at least one other architecture and, ideally, self-supervised representations closer to the state of the art are essential to support the usefulness of the method.


[A] Maxime Oquab, Timothée Darcet, Theo Moutakanni, Huy V. Vo et al., DINOv2: Learning robust visual features without supervision. In TMLR, 2024.

**Requested Changes:**

### 1. Evaluation:

Please expand the experiments by comparing against more architectures, stronger self-supervised models (e.g., DINOv2), and add some qualitative analysis (such as PCA visualizations).


### 2. Motivation and task:

Clarify why self-supervised fine-tuning on the target domain is needed, address the conflict between continual learning and pre-training, explain what is meant by the “continuous” aspect, and explain how the method aims to generalize across domains.


### 3. Writing and formatting:

Improve the overall structure/formatting of the paper, fix alignment issues with figures and tables, clean up the references for consistency, and try to make Figure 1 easier to follow.

---

> ### Author Response · Authors · 2025-09-12
> **Response to Reviewer KXij**
>
> We sincerely thank Reviewer KXij for carefully re-reading our submission and for the constructive feedback. We greatly appreciate the time invested and take these concerns seriously. Below, we address each major point raised and outline clarifications and improvements we will include in a revised manuscript.
>
> 1. **Evaluation:**( more architectures, stronger SSL backbones):
> We already compare GLARE on top of multiple modern SSL backbones (DINO, FLSL, UDI) and report results on several benchmarks, including the LoveDA dataset. For example:
>
> | Init. | Cont. pre-training | Original | HPT (31M)  | UniAdapter (14M) |
> | ----- | ------------------ |  -------- | ---- | ---------- |
> | DINO  | → DINO             | 50.3     | 28.7 | 28.4       |
> |       | → GLARE            |          |      | **50.6**   |
> | FLSL  | → FLSL             |  50.3     | 50.2 | **50.6**   |
> |       | → GLARE            |          |      | 50.5       |
> | UDI   | → UDI              | 50.9     | 50.4 | 51.1       |
> |       | → GLARE            |         |      | **51.5**   |
>
> Regarding DINOv2, we attempted experiments but were unable to produce consistent results. Specifically, once the original DINOv2 backbone is unfrozen to get its baseline performance on out-of-domain data (LoveDA), the performance drops drastically. This prevents a fair comparison with our other baselines. For this reason, we chose not to include DINOv2 or DINOv2 → GLARE in the current manuscript. We will include DINOv2 once we find a solution which perserves stability.
>
> In addition, we will expand the evaluation in the revised version with PCA visualizations of learned features to illustrate representational improvements. We also note that we already provide some qualitative analyses in Appendix D (Figure 4), where we show attention maps across different backbones.
>
> 2. **Motivation and Task**:
>
> - Our work motivates continual self-supervised pre-training as a domain-adaptation step that reuses and refines representations learned by a foundation SSL model on target-domain unlabeled data. In real-world scenarios, unlabeled data are typically abundant, while labeled data are scarce, especially on specific domains. Our goal is to leverage existing SSL models together with unlabeled target-domain data to adapt features before downstream fine-tuning.
> - Continual learning usually refers to adapting a model trained for a specific downstream task to a new domain. By contrast, continual pre-training begins with a self-supervised pre-trained model and continues unsupervised pre-training on new data, without access to labels. This makes our setting fundamentally different from standard continual learning.
> - The primary goal of our method is not to generalize across arbitrary domains, but to adapt a generalist SSL backbone to a specific target domain. Nevertheless, we do examine generalization effects in Section 6.4, where we study how much the model forgets after continual pre-training on different datasets. We observe that continual pre-training preserves performance on other datasets, and in some cases even improves it, despite being adapted to a different dataset.
>
>
> 3. **Writing and Formatting**: We thank the reviewer for this feedback. We will revise the manuscript to: improve the clarity of the presentation, fix formatting and alignment issues in figures and tables, and ensure references are consistent and cleanly formatted.
>
>
> **Final note**:
> We want to mention as well that we do not ignore comments from previous reviews. In fact, we incorporated many of the major concerns raised in earlier feedback into this submission (e.g., expanded baselines, ablations). We will do the same with the new round of reviews, ensuring that the revised version addresses the points raised here in detail to strengthen our work.

---

### Decision · Action_Editor_rVGm · 2025-12-01

**Recommendation:** Accept with minor revision

**Additional Comments:**

There are three reviews in total: two reviewers recommend “Leaning Accept” and one recommends “Leaning Reject.” I carefully read the paper and all reviews, as well as the author rebuttals and added experiments. While I acknowledge the concerns raised by the rejecting reviewer, I find that (i) these issues are either addressed to a reasonable extent in the revision, or (ii) are calibration/positioning issues rather than methodological flaws warranting rejection for TMLR. Given the overall technical soundness, minor empirical improvements, and relevance to TMLR’s audience, I do not consider the single “Leaning Reject” sufficient grounds to recommend rejection, and thus I support acceptance.

However, the paper should:

- Further clarify the distinction between continual pre-training and standard SSL or continual learning, and sharpen the motivation for domain-specific continual SSL on unlabeled data in the main text.
- Carefully position the work relative to prior parameter-efficient adaptation methods (adapters, LoRA, prompt tuning).
- Integrate all additional experiments promised and reported in the review responses.
- Improve writing, figures, and formatting, and calibrate claims so that they match the evidence (e.g., “consistent, parameter-efficient improvements” rather than strong SOTA claims).

Subject to these revisions, the paper is a good fit for TMLR and should be accepted.

**Audience:**

Yes

**Audience Explanation:**

The paper is at the intersection of self-supervised learning, vision foundation models, and domain adaptation for dense prediction. All of these topics are an interest to the TMLR audience. These sub-communities will likely find the ideas, design choices (global–regional–local consistency), and empirical observations useful

**Claims And Evidence:**

Yes

**Claims Explanation:**

The main claims are that (i) GLARE provides a continual self-supervised pre-training scheme that can adapt SSL backbones to new domains for semantic segmentation, and (ii) this can be done in a parameter-efficient way while yielding consistent, if modest, performance gains across benchmarks. After the revision and rebuttal, I believe these claims are sufficiently supported for TMLR standards:

- The method is clearly specified, and the authors provide results across multiple datasets and multiple SSL backbones, including comparisons to alternative continual pre-training strategies.
- They added from-scratch baselines and extra experiments on classification datasets (Derm7pt, COVIDx).
- The improvements are modest in absolute terms but the gains are consistent and align with the claims made in the paper.

---

> ### Author Response · Authors · 2025-12-30
> **Thank you & Camera-ready version uploaded**
>
> Dear Ozan,
>
> Thank you for your careful review and thoughtful recommendation.
>
> We have uploaded the camera-ready version of our paper, incorporating the requested revisions, including clarifications on continual SSL, positioning relative to parameter-efficient methods, integration of additional experiments, and improvements to writing and presentation.
>
> We appreciate the time and care you devoted to overseeing the review process.
>
> Best regards